# Immunological and Clinical Impact of DAA-Mediated HCV Eradication in a Cohort of HIV/HCV Coinfected Patients: Monocentric Italian Experience

**DOI:** 10.3390/diagnostics11122336

**Published:** 2021-12-11

**Authors:** Andrea Marino, Gabriella Zafarana, Manuela Ceccarelli, Federica Cosentino, Vittoria Moscatt, Gabriele Bruno, Roberto Bruno, Francesco Benanti, Bruno Cacopardo, Benedetto Maurizio Celesia

**Affiliations:** 1Unit of Infectious Diseases, Department of Clinical and Experimental Medicine, ARNAS Garibaldi Nesima Hospital, University of Catania, 95123 Catania, Italy; andreamarino9103@gmail.com (A.M.); gabryzafarana@yahoo.it (G.Z.); Manuela.ceccarelli86@gmail.com (M.C.); federicacosentino91@gmail.com (F.C.); vittoria.moscatt@gmail.com (V.M.); bruga90@gmail.com (G.B.); Bruno_roberto@virgilio.it (R.B.); Francesco.benanti61@gmail.com (F.B.); cacopard@unict.it (B.C.); 2Unit of Infectious Diseases, Department of Clinical and Experimental Medicine, University of Messina, 98100 Messina, Italy

**Keywords:** HIV, HIV/HCV coinfections, HCV treatment

## Abstract

HCV treatment became available for all infected patients regardless of their comorbidities, especially for HIV coinfected subjects, leading to an improvement in both clinical and immunological conditions. We retrospectively analyzed a cohort of HIV/HCV coinfected patients treated with DAA therapies; data regarding epidemiological, viral-immunological, and hepatic parameters before and after DAA administration have been collected. Drug-drug interactions between DAA and both antiretroviral therapy and non-ART-drugs were also evaluated; the study showed the efficacy of DAA schedules in HCV eradication also for HIV/HCV patients with multiple comorbidities and assuming many different drugs. Principal issues are still represented by drug interactions, pill burden, and patients’ compliance. These concerns have to be taken into account, especially in HIV patients for whom the immunological state and ART interactions should always be considered.

## 1. Introduction

Worldwide, Hepatitis C Virus (HCV) infection accounts for a significant number of chronic liver diseases in People Living With Human Immunodeficiency Virus (PLWHIV). Patients coinfected with HCV and HIV experience a faster liver disease progression than mono-infected individuals, with a higher risk of hepatic decompensations, liver failure, and hepatocellular carcinomas (HCC) [1,2]. In addition, HIV infection leads to a series of unfavorable events such as augmented HCV replication, hepatic inflammation, hepatocyte apoptosis, and microbial translocation, resulting in the deterioration of the specific immune responses against HCV [3,4]. Fortunately, the natural history of chronic Hepatitis C infection has dramatically changed since the introduction of Direct Antiviral Agents (DAA) regimens. In clinical settings, these regimens have led to sustained virological response at 12 weeks (SVR12) after the end of treatment (EOT) in up to 80–96% of cases. DAA response in HIV-HCV coinfected individuals is similar to HCV mono-infected ones. The cure of HCV infection reduces the risk of liver cancer by 76% and the risk of death by 50%; theoretically, it could also reduce HCV transmission [5]. Consequently, a DAA based therapy is recommended as standard therapy regardless of HIV coinfection and antiretroviral therapy (ART) [5,6]. We retrospectively analyze the data of a monocentric cohort of HIV/HCV coinfected patients treated with DAAs to evaluate the impact of eradicative therapy on liver fibrosis and on immunological and virological markers of HIV infection.

## 2. Materials and Methods

This is a retrospective observational monocentric cohort study. Data were collected from medical records and updated until 1 January 2021. All patients signed an informed consent to collect and anonymously analyze the data.

### 2.1. Primary Endpoints

-To evaluate the real world DAA efficacy in a cohort of patients HIV/HCV coinfected.-To evaluate epidemiological and clinical characteristics of the same cohort.-To define the strategies driving the choice of DAA treatment in this population.-To assess Drug-Drug Interactions (DDIs) for any DAA regimen, regardless of HCV different genotypes.

### 2.2. Secondary Endpoints

To evaluate the impact of DAA-mediated HCV eradication on the following immunological and biochemical parameters in PLWHIV:-CD4+, CD8+ cells count, and CD4/CD8 ratio;-Liver fibrosis non-invasive scores: Aspartate Aminotransferase to Platelet Ratio Index (APRI) [7] and Fibrosis-4 (FIB-4) [8];-Total cholesterol levels, high-density lipoprotein (HDL)-cholesterol levels, and triglycerides levels.

### 2.3. Enrollment

All the HIV-positive patients with chronic HCV infection who started DAA-mediated HCV eradication therapy at the Infectious Diseases Department at Garibaldi Nesima Hospital (Catania) were considered eligible for the study.

### 2.4. Inclusion Criteria

HIV-RNA positive patients with chronic hepatitis C treated with direct-acting antiviral agents.

### 2.5. Exclusion Criteria

-HIV/HCV coinfected patients responding to IFN-based eradication therapy;-HIV/HCV coinfected patients who were lost to follow-up, died, or transferred before receiving any anti-HCV treatment;-HIV/HCV coinfected patients whose HCV infection resolved spontaneously.

### 2.6. Methods

The start of DAA therapy was considered the baseline for the study.

The following parameters were collected from medical records:-Epidemiological characteristics: sex, age, ethnic group, and HIV and HCV transmission risk factors (IDU, MSM, others);-Clinical parameters: date of HIV diagnosis; date of HCV diagnosis; HCV genotype; HCV RNA (IU/mL); HIV-RNA (copies/mL); CD4+ T-cells count (cells/mm^3^), CD8+ T-cells count (cells/mm^3^), and CD4/CD8 ratio; AST (UI/L); ALT (UI/L); platelets count (10^3^/mm^3^); hepatitis B serology (HBcAb positivity); previous failure to HCV eradication therapy; DAA starting date; comorbidities; ART drugs; and domiciliary therapy assumed. HIV RNA and HCV RNA have been quantified by RT-PCR technique (Roche Diagnostic).

T lymphocyte subpopulations have been studied by BD FACSCanto Flow Cytometer.

-Liver fibrosis, indirectly evaluated with hepatic elastography (FibroScan 502 Touch – Echosens) [9];-DAA therapy administered: ledipasvir/sofosbuvir (SOF/LED), sofosbuvir/daclatasvir (SOF/DAC), elbasvir/grazoprevir (ELB/GRZ), ombitasvir/paritaprevir/ritonavir ± dasabuvir (3D), velpatasvir/sofosbuvir (VEL/SOF), and glecaprevir/pibrentasvir (GLE/PBR).-Efficacy of DAA therapy was evaluated with the achievement of Sustained Virological Response (SVR12), defined as undetectable plasma HCV RNA (<15 IU/mL) at least 12 weeks after the end of anti-HCV treatment. In addition, we used the term EOT (End Of Therapy) to define undetectable plasma HCV RNA at the end of the therapy.

DDIs with drugs regularly assumed by HIV/HCV co-infected patients were assessed using www.hep-druginteractions.org (Last accessed on 14 April 2021) [10]. Interactions between both DAAs and HIV therapies and DAAs and drugs used for other comorbidities were considered separately.

DDIs were assigned to four different risk categories:-Green: no clinically significant interactions expected;-Yellow: potential weak interaction (additional monitoring, dose adjustment, or therapy adjustment not required);-Amber: potential significant interaction (additional monitoring, dose adjustment, or therapy adjustment required);-Red: co-administration not recommended or contraindicated.

To evaluate the clinical and immunological responses after 48 weeks from the end of DAA therapy (SVR48), the following clinical and laboratory parameters were collected:

-CD4+ T-cells count (cells/mm^3^), CD8+ T-cells count (cells/mm^3^), and CD4/CD8 ratio; ALT (UI/L); AST (UI/L); platelet count (10^3^/mm^3^), total cholesterol levels (mg/dL), and high density lipoprotein (HDL)-cholesterol levels (mg/dL); and triglycerides levels (mg/dL).

To better define the CD4+ and CD8+ T-cells counts, and CD4/CD8 ratio trends, an additional measurement of these parameters was collected six months after EOT.

The liver fibrosis non-invasive serum markers Aspartate Aminotransferase to Platelet Ratio Index (APRI) and Fibrosis-4 (FIB-4) were calculated at the baseline and at SVR48, using these formulas:-FIB-4: Age ([year] × AST [U/L])/((PLT [10(9)/L]) × (ALT [U/L]);-APRI: [(AST/upper limit of the normal AST range) × 100]/Platelet Count.

### 2.7. Statistical Analysis

The qualitative variables were described with frequencies and percentages, while the quantitative variables with mean, median, standard deviation, minimum, maximum, and percentiles.

Inferential analysis was performed with Graphpad Prism 9.0.2 (GraphPad Software, San Diego, CA, USA) for MacOS. Variables distribution was tested with Kolmogorov–Smirnov’s test.

In the evaluation of the parameters not normally distributed, non-parametric tests (Friedman’s, Wilcoxon) were applied. In the case of variables normally distributed, parametric tests (*t*-test, ANOVA, chi-square) were applied to test any statistically significant differences.

## 3. Results

### 3.1. Epidemiological Characteristics

Overall, out of 670 HIV positive patients followed at the Infectious Disease Department at Garibaldi Nesima Hospital in Catania, 115 patients tested positive for the HCV antibody. Twenty-two out of them were HCV RNA undetectable with no history of previous treatment and were considered to have cleared C virus spontaneously; 10 patients were responders to PEG-IFN/RBV eradication treatment; 18 died or were lost to follow-up or transferred without starting any anti-HCV treatment. Finally, 65 HIV/HCV patients starting DAA therapy were considered eligible.

Comprehensively, 18 (28%) patients were female and 47 (72%) were male. The median age was 52 years (IQR 49–56). The HIV and HCV main risk factors were intravenous drug use (78%), followed by unprotected MSM intercourse (12%), and blood transfusion (5%).

Forty-two (65%) patients were HCV genotype 1, 16 (25%) were HCV genotype 3, 6 (9%) were genotype 4, and 1 (1%) was genotype 2. At the time of DAA initiation, the median HCV RNA was 6.5 (IQR 6–7) log10 IU/mL. Finally, 21 patients (32%) had markers of previous HBV infection (positive HBcAb). As far as HIV status, the HIV-RNA was ≤50 copies/mL in 56 (86%) patients. The CD4+ T cell count was >500 cells/mm^3^ in 43 (66%), between 351 and 500 cells/mm^3^ in 8 (12%), and ≤350 cells/mm^3^ in 11 (17%) patients. Significant fibrosis, defined as Metavir stages stratified by fibroscan F2 and F3, was observed in 17 patients (26%). Advanced liver fibrosis (F4) was found in 14 patients (21%) (Table 1).

Regarding viral eradication, 59 patients of the cohort (91%) reached SVR12. The remaining 6 patients were lost to follow-up. The percentage of comorbidities is shown in Figure 1.

### 3.2. DAAs Treatment and Non-ART Drug-Drug Interactions

With reference to DAA treatments, 29 (45%) patients took VEL/SOF, 12 (18%) took GLE/PBR, 7 patients (11%) took SOF/LED, 6 (9%) patients took SOF/DAC, 6 (9%) patients took 3D, 4 (6%) patients took ELB/GRZ, and 1 patient took sofosbuvir and ribavirin.

At the time of DAA initiation, 29 (45%) patients had at least one comorbidity, 18 (28%) patients reported one comorbidity, and 11 (17%) patients reported at least two comorbidities (Figure 2a).

Not considering ARV medications, 19 (29%) patients did not assume any other drug at baseline. Twenty-six (40%) patients assumed 1 or 2 non-ARV drugs. Twenty (31%) patients assumed ≥3 non-ARV drugs (Figure 2b).

Overall, 62 different substances were found as outpatient medications (TARV not included). Cholecalciferol (21.5%), aspirin (9%), clonazepam (8%), ursodeoxycholic acid (8%), and methadone (6%) were the most frequently used drugs.

For Drug-Drug Interactions, the results are shown in Figure 3.

Proton pump inhibitors and some antidyslipidemic drugs represented the main categories at high risk of interaction with DAA regimens. ELB/GRZ revealed the lowest DDI risk (5%), whereas 3D showed potentially significant DDIs for 18% of the chronic treatment. Significant DDIs were expected for 11% of patients using both SOF/LDV and SOF/VEL. DDIs with SOF/DAC and GLE/PBR are expected in 8% of patients.

With respect to contraindications, 3D and SOF/LDV had a risk of 3% and 2%, respectively.

### 3.3. ART Schedules and ART-Drug-Drug Interactions

The nucleoside reverse transcriptase inhibitors (NRTI) backbone plus integrase inhibitors (INSTI) was the most represented ART regimen (44%), followed by NRTI backbone plus non-nucleoside reverse transcriptase inhibitors (NNRTI) (17%), and NRTI backbone plus protease inhibitors (PI) (14%).

As regards NRTI backbone, 51% received emtricitabine (FTC) plus tenofovir alafenamide (TAF), 41% received emtricitabine (FTC) and tenofovir disoproxil fumarate (TDF), and 8% abacavir (ABC) and lamivudine (3TC).

The majority of patients preventively modified ARV treatment in order to minimize DDIs with DAA regimens.

### 3.4. DAA-Mediated HCV Eradication Effect on CD4+ and CD8+ T Lymphocyte

Only patients with at least one measurement of CD4+ and CD8+ count at DAA initiation (BL), one measurement six months after the EOT (T1), and one further measurement at SVR48 (T2) were included. The study population included 52 patients.

The ANOVA test showed no statistically significant differences between CD4+ count values at BL, those at T1, and those at T2 (Figure 4a).

Friedman’s test showed no statistically significant differences between CD8+ count values at BL, those at T1, and those at T2 (Figure 4b).

As regards CD4/CD8 ratio, a statistically significant difference was found between the values at BL and those at T2 (*p* = 0.021) (Figure 4c).

### 3.5. Dynamics of APRI and FIB-4

After DAA treatment, the median FIB-4 decreased from 1.81 to 1.66 (*p* = 0.46), and the median APRI decreased from 0.59 to 0.36 (*p* < 0.0001) (Figure 5).

### 3.6. Lipid Profile Changes

The paired samples *t*-test showed statistically significant differences between total cholesterol values at baseline (BL) and those at SVR48 (*p* = 0.002) (Figure 6a).

The Wilcoxon test showed a statistically significant difference between HDL-cholesterol levels at BL and SVR48 (*p* = 0.01) (Figure 6b). No statically significant difference was found between triglycerides levels at BL and SVR48 (Figure 6c).

## 4. Discussion

With the widespread use of antiretroviral therapy (ART), AIDS-related mortality has declined sharply and life expectancy of PLWH has significantly increased [11,12,13].

Therefore, PLWH chronic infections, such as hepatitis C, have gained even more importance as possible causes of mortality and morbidity [14].

Nowadays, direct-acting antiviral (DAA) treatment leads to HCV cure, leading to a sustained virologic response (SVR) in more than 95% of HCV patients, including those with HIV/HCV coinfection [15].

Reinforced by the potential impact of DAAs, the World Health Organization (WHO) has endorsed the elimination of HCV as a public health threat by 2030 [16].

To strive for this ambitious goal, the single global challenge can be broken into smaller and more easily achievable treatment and prevention targets for subpopulations, such as HIV/HCV coinfected patients [17].

For those patients, the coinfection might add clinical obstacles to HCV treatment, such as delays in the initiation, potential drug-drug interactions, and an increased pill burden [18].

Several studies have shown that SVR is associated with a decreased risk of hepatocellular carcinoma (HCC) and an overall decline in liver-related mortality and morbidity [19].

The management of HIV/HCV coinfected patients should always include preventive liver disease measures (i.e., counseling about alcohol consumption, counseling about the possible risk of HCV reinfection after cure, the avoidance of hepatotoxic antiretroviral drugs, or access to HCC surveillance program, depending on the individual risk).

It is known that liver-related outcomes among HIV/HCV infected patients are worse in the case of the triple HIV/HBV/HCV coinfection [20]. Interestingly, recent evidence shows a significantly increased risk of liver damage also in HIV/HCV coinfected people if signs of resolved HBV infection (HBcAb positivity) are present, suggesting the need for a careful HBV serology evaluation [21,22].

The HIV/HCV coinfected group is an important part of all PLWH population in Italy (in ICONA study cohort, HIV/HCV coinfected patients are 26.3%) and, in recent years, a decrease in coinfected patients among the newly diagnosed HIV positive has been observed [23].

This study aimed to examine the epidemiological features, the treatment profiles, and the viral eradication outcomes in PLWHIV with chronic hepatitis C.

In our cohort, the main risk factors for HIV and HCV acquisition were intravenous drug use along with unprotected intercourse reported by men who have sex with men (MSM). These data are consistent with the current epidemiological Italian scenario [24].

According to epidemiological data, IDUs are still the group at highest risk of new HCV infection in Italy, while sexual transmission stands out as a common mode of acquisition among MSM, especially if they have other risk factors, such as HIV.

The introduction of blood-screening for HCV in the early 1990s strongly decreased HCV transmission from blood transfusions. In the study cohort, 5% of patients had a history of blood transfusions as a risk factor.

Regarding DAA efficacy, 97% of the patients presented undetectable HCV RNA at the end of treatment (EOT). Furthermore, 3% of patients started the treatment, but they were lost to follow-up before EOT measurement.

The main drive for the decision of the appropriate DAA regimen, according to guidelines [5], was HCV genotype and the fibrosis stage. However, having multiple options, we select the regimen on the basis of potential DDIs, patients’ preferences about treatment duration, and pill burden.

Potential drug-drug interactions are among the main factors, together with HCV genotype and the stage of liver fibrosis, that may influence the choice of the DAA regimen.

The significant number of comorbidities and polypharmacy in HIV/HCV coinfected patients increases the risk for serious DDIs during DAA treatment. In the management of the HIV/HCV coinfected patient, any changes in ART should be introduced before DAA initiation. In our cohort, the majority of patients preventively modified ART treatment to adapt it to DAA therapy.

Zuckerman et al. [25] reported that the most common change in HIV/HCV coinfected population is adjusting from TDF to tenofovir alafenamide (TAF)-based regimens. As more HIV/HCV coinfected patients are treated with TAF-based therapies, the need to change ART to start DAA-based treatment may decrease.

Concerning the analysis of the potential DDIs with all outpatient medications, our study showed that the most used DAAs regimens have a lower risk of DDIs (5% for ELB/GRZ, 8% for GLE/PBR, 11% for SOF/VEL), whereas, for the 3D regimen (ombitasvir/paritaprevir/ritonavir ± dasabuvir), potentially significant DDIs could be expected for 18% of patients.

HCV infection has been described as a contributor to poor immunological recovery in PLWH. However, a recent study in the context of the ICONA and HEPA-ICONA observational cohorts showed that HCV eradication following DAA treatment does not have an impact on CD4+ T cell recovery in PLWHIV [26].

To clarify the impact that HCV eradication might have on immune activation in PLWH, we evaluated CD4+, CD8+ cell count, and CD4/CD8 ratio prior- and post-HCV eradication, revealing that DAA treatment did not show a beneficial effect on CD4+ T cell recovery. In addition, no significant changes were detected for both the CD8+ T cell count and the CD4/CD8 ratio. Although these findings may be limited by the small sample of the study [27], some authors showed an improvement in T cells immune activation along with normalization of the T-cell phenotype following SVR due to DAA therapy [28]. A significative example is the work of Emmanuel et al. [17], which revealed how the number of activated CD4+ and CD8+ T cells declined from pre-DAA therapy to post-SVR in 60 HIV/HCV-coinfected patients. In addition, Meissner et al. [6], studying peripheral blood lymphocytes early during DAA therapy, showed an increase in total CD4^+^ and CD8^+^ T cells and, at the same time, a reduction in activated CD4^+^ and CD8^+^ T cells, with this effect being sustained up to one year after SVR; furthermore, the same study postulated a redistribution of T cells from the liver to the periphery after DAA treatment.

In our study, 17 patients (26%) had significant fibrosis at baseline, defined as Metavir stages stratified by liver elastography equal to F2 or F3. Advanced liver scarring (F4) was found in 14 patients (21%).

After treatment, the median APRI was decreased from 0.59 to 0.36 (*p* = 0.001), and the median FIB-4 was decreased from 1.81 to 1.66.

The study also investigated changes in lipid profiles in HIV/HCV coinfected patients after HCV eradication.

It is known that HCV alters lipid pathways to enhance its replication. HCV infection could cause hypocholesterolemia; therefore, successful treatment may lead to elevation of cholesterol, including LDL-cholesterol [26]. Indeed, several studies reported pro-atherogenic lipid changes in patients with SVR [29,30]. In our cohort, the serum total cholesterol (*p* = 0.02), HDL-cholesterol (*p* = 0.01), and triglycerides levels increased after HCV eradication.

However, the actual changes in the serum lipid profile and liver steatosis and their relationship with successful HCV eradication by DAAs have not been clarified. Further studies will be necessary to evaluate the possible negative changes of the lipid profile.

### Limits of the Study

Data collection for this study was performed in 2020, with the latest update on 1 January 2021. Difficulties in collecting data due to the COVID-19 pandemic were encountered.

The follow-up of some patients was affected by delays related to the pandemic and some data gaps in medical records (which led to the exclusion of patients from the analyses) may have been caused by this fact.

The sample size may have affected statistical analyses concerning changes in clinical parameters in the post-treatment setting. In addition, the study population is quite heterogeneous regarding comorbidities, stage of liver fibrosis, and outpatient drugs, and these factors were not considered in the evaluation of lipid changes and hepatic fibrosis.

## 5. Conclusions

This retrospective observational study confirmed the efficacy of DAA treatment in PLWH with chronic hepatitis C. The study also underlined the need for accuracy and attention in the management of these patients.

HIV/HCV coinfected patients often have multiple comorbidities and are treated with various drugs, in addition to ART. A detailed preventive check for all drugs is mandatory for all patients before starting direct-acting antiviral therapy and at every control visit. A careful evaluation of potential drug-drug interactions is essential to prevent any adverse events. The most used and feasible DAAs regimens have a lower risk of DDIs compared to the old regimen 3D.

According to our data, DAA treatment in HCV/HIV patients does not show an effect on CD4+ T or CD8+ T in the months following treatment. Regarding the CD4/CD8 ratio, a statistically significant increase was found between the values at BL and those at T2 (*p* = 0.02). A longer follow-up and a larger sample size might be required to highlight more significant changes in the immunological assessment after DAA treatment.

## Figures and Tables

**Figure 1 diagnostics-11-02336-f001:**
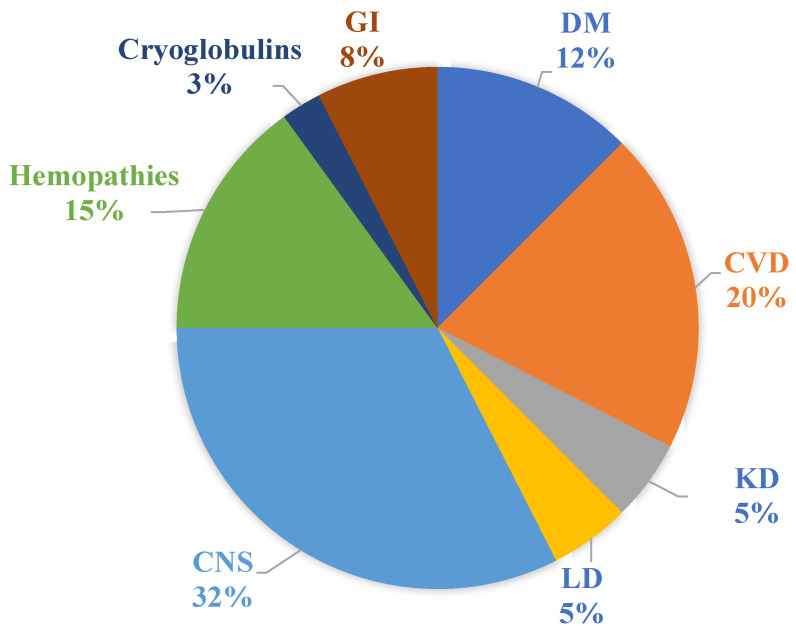
Percentage of comorbidities of the cohort. DM: diabetes mellitus; CVD: cardiovascular diseases; KD: kidney diseases; LD: lung diseases; CNS: central nervous system diseases; GI: gastrointestinal diseases.

**Figure 2 diagnostics-11-02336-f002:**
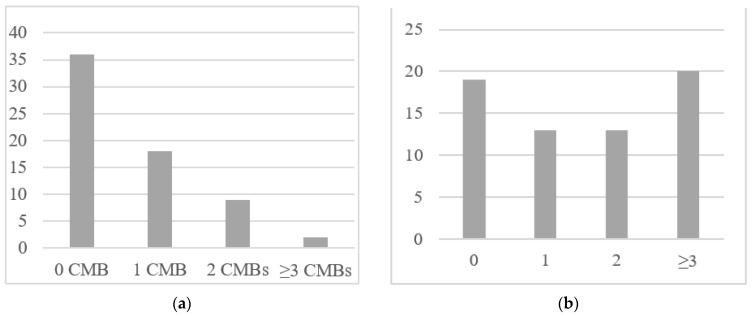
(**a**) Frequency of comorbidities (CMB) in the cohort of patients. (**b**) Frequency of number of drugs regularly assumed by HCV/HIV co-infected patients (ART not included).

**Figure 3 diagnostics-11-02336-f003:**
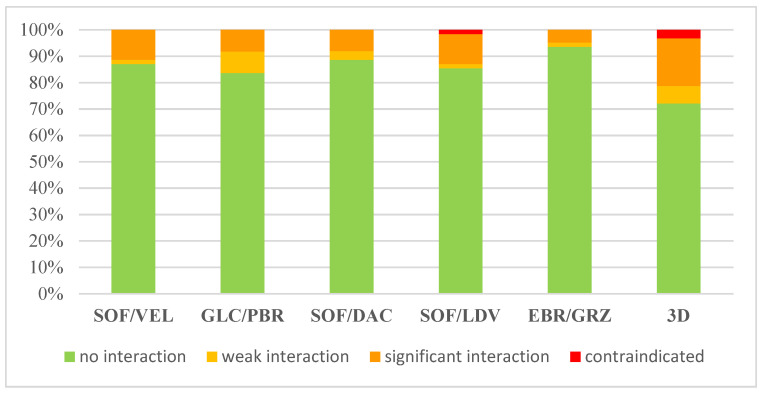
Expected drug-drug interactions between outpatient medications and DAA (ART not included).

**Figure 4 diagnostics-11-02336-f004:**
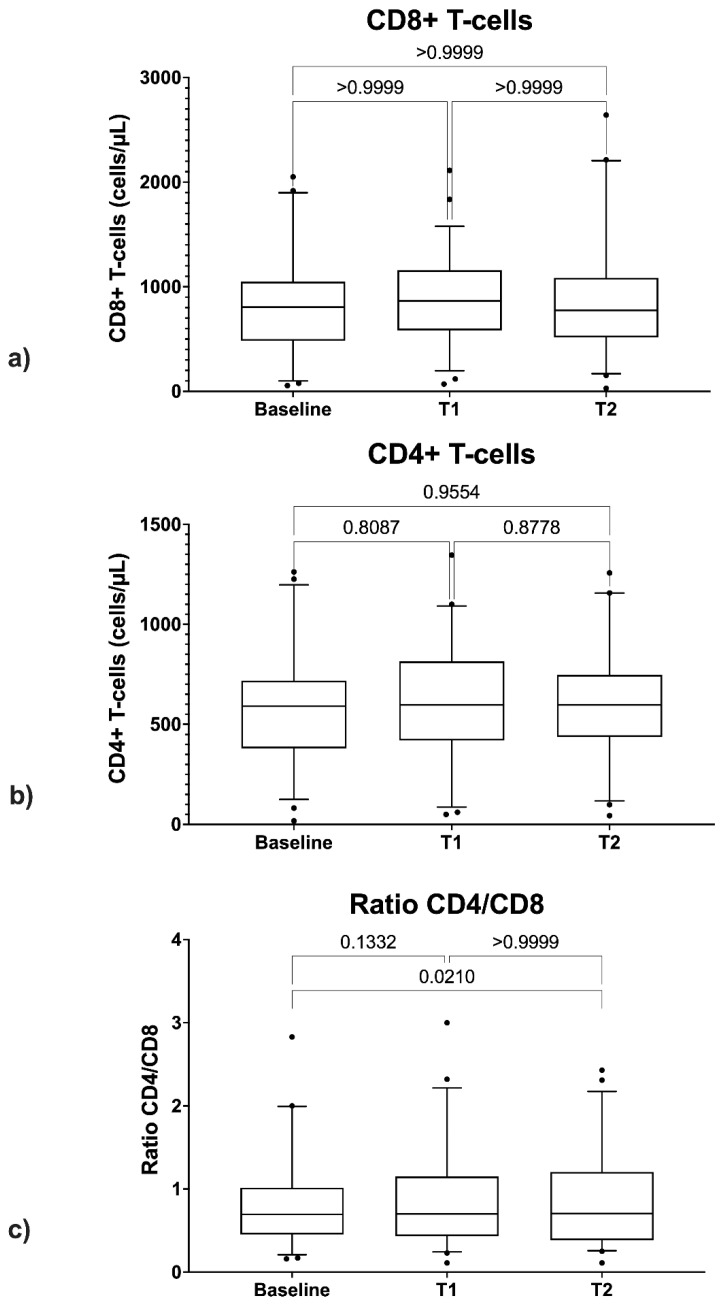
Comparison between CD4+ cells (**a**), CD8+ cells (**b**), and CD4/CD8 ratio (**c**) at baseline, EOT (T1), and SVR48 (T2).

**Figure 5 diagnostics-11-02336-f005:**
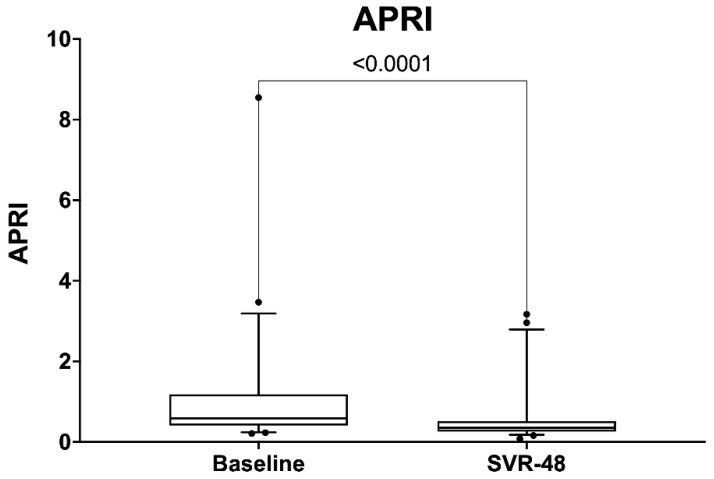
Comparison between APRI score at baseline and SVR48.

**Figure 6 diagnostics-11-02336-f006:**
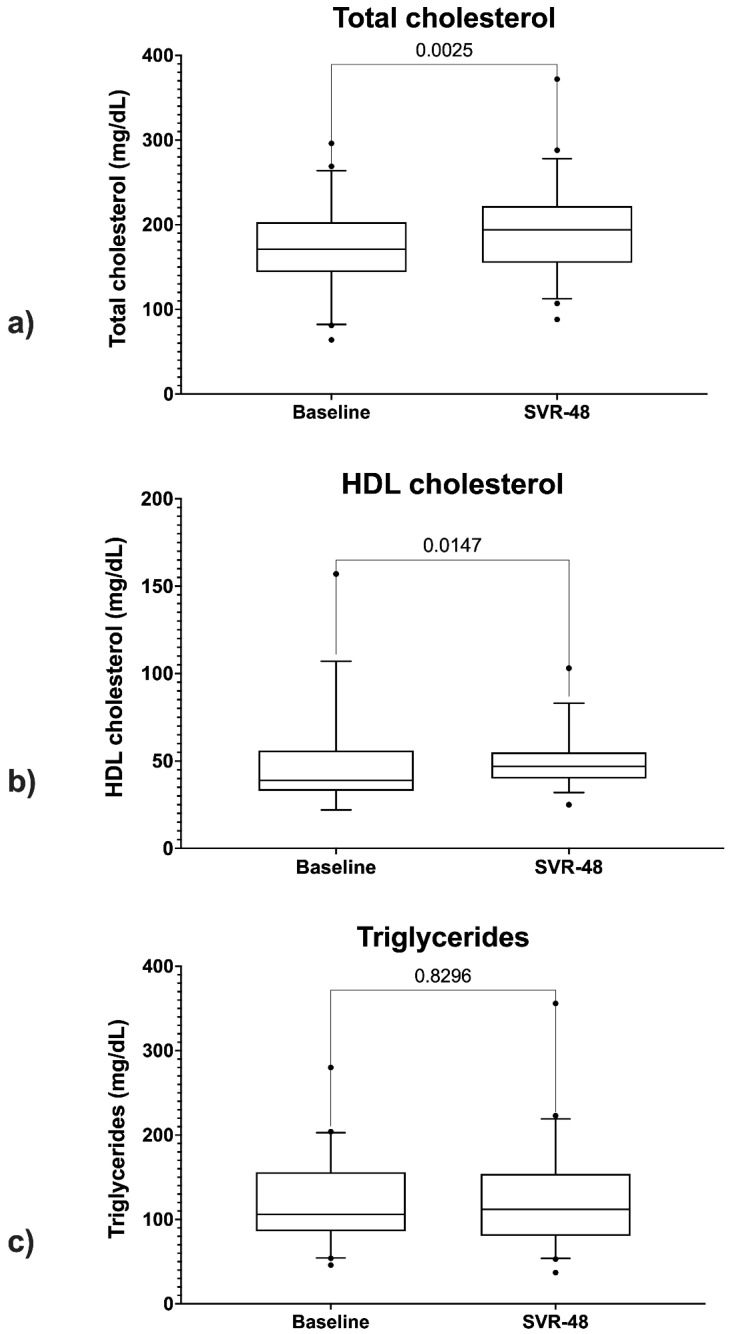
Comparison between total cholesterol levels (**a**), HDL levels (**b**), and triglycerides (**c**) at baseline and SVR48.

**Table 1 diagnostics-11-02336-t001:** Clinical and epidemiological characteristics of the whole cohort.

Characteristics		Population (*n* = 65)	(%)
Gender, *n* (%)	F	18	28%
	M	47	72%
Italian nationality	Yes	64	98%
	No	1	2%
Age, years, median IQR	52 (49–56)		
Year of HIV diagnosis, mode	1985		
Year of HCV diagnosis, mode	1993		
HCV-RNA, log10, median (IQR)	6.5 (6–7)		
HBcAb positivity *n* (%)	21 (32%)		
IFN-experienced, *n* (%)	11 (17%)		
Risk factors	IDU	51	78%
	MSM	8	12%
	Transfusions	3	5%
	Unknown	3	5%
CD4+, cell/mm^3^, median (IQR)	642 (464–870)		
HIV-RNA, log10, median (IQR)	1.30 (1.25–1.36)		
Baseline HIV-RNA levels, copies/mL	≤50	56	86%
	50–400	7	11%
	not available	2	3%
HCV genotype, n (%)	1	42	65%
	2	1	1%
	3	16	25%
	4	6	9%
METAVIR Stage Stratified with liver elastography (Fibroscan), *n* (%)	F1	31	48%
	F2	13	20%
	F3	4	6%
	F4	14	21%
	not available	3	5%

## Data Availability

The data presented in this study are available on request from the corresponding author. The data are not publicly available due to privacy policy of the Hospital and Department involved in the study.

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
