# Peer review of "Immunological and Clinical Impact of DAA-Mediated HCV Eradication in a Cohort of HIV/HCV Coinfected Patients: Monocentric Italian Experience"

_diagnostics, 2021, doi:10.3390/diagnostics11122336_

Round 1

Reviewer 1 Report

The work presented by Marino et al ., on the immunological and clinical impact of DAA-mediated HCV eradication in a cohort of HIV / HCV co-infected patient is interesting and well structured.
Before publication, I recommend some revision also to make the text easier to read.
In general there are many acronyms so I would recommend a small paragraph at the end of the text where you can report them all.
In particular:
Introduction
lines 36 and 42: check spaces before bibliographic notes;

materials and methods 

If you use the bulleted list as in point 2.1, use the same style in all the other lists 2.2, 2.5, ect

line 88: check the space before the bibliography; 

line 97: put the site in the bibliographic notes,
lines 108-110: after the comma put the lowercase character;

Results

line 140: the meaning of MSM is missing.

Figure 1 is not mentioned in the text, if you want to keep adjusting the style, for example the% in KD is missing. Figure 2: It should be moved under the corresponding text, a title should be placed and authors should check the style suggested by the MDPI guidelines for images and graphics. Figure 3: Move below the corresponding text. Figure 5: should be renamed, because there is no figure 4, in the text at lines 198, 200 and 202. The graphs are not seen well. Figure 6: Should be replaced with a better quality image and renamed as 5, in line 208 text. Figure 7: like the others, the graphics should be replaced, of the same size and under the corresponding text. This too must be renamed, in the text lines 214, 216 and 217. Discussion:  lines 229, 230: bibliography missing
line 305: Check for spaces before parentheses
line 310: I don't think it's necessary to write the limits paragraph in uppercase and bolt.
References
References should be set to 30, and in the style suggested by MDPI.

Author Response

In general there are many acronyms so I would recommend a small paragraph at the end of the text where you can report them all.
Reply: Thank you for the precious and valuable suggestion. We added the “Abbreviations” paragraph as suggested.

In particular:
Introduction
lines 36 and 42: check spaces before bibliographic notes;

Reply: We fixed the typo

materials and methods 

If you use the bulleted list as in point 2.1, use the same style in all the other lists 2.2, 2.5, ect

Reply: We modified the style.

line 88: check the space before the bibliography; 

Reply: we fixed the typo

line 97: put the site in the bibliographic notes

Reply: Thank you for the suggestion, we added the site in the references section.

lines 108-110: after the comma put the lowercase character;

Reply: We fixed the typo

Results

line 140: the meaning of MSM is missing.

Reply: We clarified every abbreviation in the paragraph, as you suggested.

Figure 1 is not mentioned in the text, if you want to keep adjusting the style, for example the% in KD is missing. Figure 2: It should be moved under the corresponding text, a title should be placed and authors should check the style suggested by the MDPI guidelines for images and graphics. Figure 3: Move below the corresponding text. Figure 5: should be renamed, because there is no figure 4, in the text at lines 198, 200 and 202. The graphs are not seen well. Figure 6: Should be replaced with a better quality image and renamed as 5, in line 208 text. Figure 7: like the others, the graphics should be replaced, of the same size and under the corresponding text. This too must be renamed, in the text lines 214, 216 and 217. 

Reply: We are sorry for the mistakes we made with Figures. We tried to fix them all. Thank you for your valuable consideration.

Discussion:  lines 229, 230: bibliography missing

Reply: We added corresponding references.

line 305: Check for spaces before parentheses

Reply: We fixed the typos

line 310: I don't think it's necessary to write the limits paragraph in uppercase and bolt.

Reply: Thank you for the suggestion. We modified the font.

References
References should be set to 30, and in the style suggested by MDPI.

Reply: Thank you for the precious suggestion. We added and discussed other references according to other reviewers.

Reviewer 2 Report

The article by Marino et.al " Immunological and clinical impact of DAA- mediated HCV eradication in a cohort of HIV/HCV coinfected patients: monocentric Italian experience"  is a matter of higher interest and greater priority. I have following comments:

  1. This manuscript describes no immunological changes were observed post DAA treatment based on total CD4 and CD8 T cell counts, however no T cell phenotypes and HCV specific T cell functions were measured. However, there are several studies that report immunological recovery after DAA treatment in HIV/HCV co-infected patients. Please comment on this.
  2. Figures need to be improved for better visualization. Fonts are very small and not clearly visible.
  3. References need to be added : lines (258-260) and lines (261-262).
  4. Methods need to be more descriptive and clear.
  5. Please comment on how HCV genotypes and stage of liver fibrosis influence the treatment option on the based on this study. 
  6. In line 287 author stated that "To clarify the impact that HCV eradication might have on immune activation" . however no immune activation (no activation markers ) was analyzed. It is better to say that "impact that HCV eradication might have on total CD4 and CD8 T cell profiles or counts".

Author Response

Reviewer 2

  1. This manuscript describes no immunological changes were observed post DAA treatment based on total CD4 and CD8 T cell counts, however no T cell phenotypes and HCV specific T cell functions were measured. However, there are several studies that report immunological recovery after DAA treatment in HIV/HCV co-infected patients. Please comment on this.

Reply: We would like to thank the precious point of view of the reviewer. We discussed the argument he had suggested adding references and debating them in discussion section.

  1. Figures need to be improved for better visualization. Fonts are very small and not clearly visible.

Reply: We tried to improve the figures and maximize the font.

  1. References need to be added: lines (258-260) and lines (261-262).

Reply: We added references as suggested.

  1. Methods need to be more descriptive and clear.

Reply: We added some references and tried to clarify some points as kindly suggested; however, we would be very grateful if the reviewer will point out what should be clarified in the method section, if it will be still necessary.

  1. Please comment on how HCV genotypes and stage of liver fibrosis influence the treatment option on the based on this study. 

Reply: Thank you for the valuable suggestion. The main drive for the choice of the DAA regimen, according to guidelines, was the genotype and the fibrosis stage. However, having multiple options, we chose the regimen on the basis of potential DDIs, patients’ preferences, and pill burden. We added some lines regarding that argument in the discussion section.

  1. In line 287 author stated that "To clarify the impact that HCV eradication might have on immune activation" . however no immune activation (no activation markers) was analyzed. It is better to say that "impact that HCV eradication might have on total CD4 and CD8 T cell profiles or counts".

Reply: Thank you for your opinion. We modified the sentence according to your suggestion.

Once again we would like to thank the editor and the reviewers for the time they spent to improve our manuscript. We hope with the changes proposed the manuscript would be accepted in your valuable journal.

Round 2

Reviewer 2 Report

My comments are :

Figure 4, 5 and 6 needs lot of improvement for the better resolution, fonts size should be increased for the axis labels as well as p values.

In Figure 4, use either lower case or upper case for T1 and T2. its different in the figure and figure legends, need to be consistent.

Author Response

Reviewer 2

Figure 4, 5 and 6 needs lot of improvement for the better resolution, fonts size should be increased for the axis labels as well as p values.

In Figure 4, use either lower case or upper case for T1 and T2. its different in the figure and figure legends, need to be consistent.

Reply: Thank you for your valuable suggestion. We modified the figures according to your recommendations.